

# Spontaneously broken (-1)-form U(1) symmetries

Daniel Aloni[1,2], Eduardo García-Valdecasas[1,3,4], Matthew Reece[1] and Motoo Suzuki[1,5]

**1** Jefferson Physical Laboratory, Harvard University, Cambridge, MA 02138, USA
**2** Physics Department, Boston University, Boston, MA 02215, USA
**3** SISSA, Via Bonomea 265, Trieste 34136, Italy
**4** INFN, Sezione di Trieste, Via Valerio 2, 34127, Italy
**5** KEK Theory Center, Tsukuba 305-0801, Japan

## Abstract

Spontaneous breaking of symmetries leads to universal phenomena. We extend this notion to $(-1)$-form U(1) symmetries. The spontaneous breaking is diagnosed by a dependence of the vacuum energy on a constant background field $\theta$, which can be probed by the topological susceptibility. This leads to a reinterpretation of the Strong CP problem as arising from a spontaneously broken instantonic symmetry in QCD. We discuss how known solutions to the problem are unified in this framework and explore some, so far *unsuccessful*, attempts to find new solutions.



## 1 Introduction and summary

Symmetries are extremely useful for understanding quantum theories. In quantum field theory, symmetries have traditionally been taken to act on local operators and to obey a group law multiplication, but recent years have seen many generalizations, starting with $p$-form generalized global symmetries [1, 2]. The subsequent literature is too large to comprehensively cite here; instead, we refer readers to several recent pedagogical reviews [3–9]. Generalized symmetries have proved to be useful for characterizing phase transitions, strong dynamics, and other nonperturbative aspects of quantum field theory. Quantum gravity, in contrast to quantum field theory, is believed to lack such symmetries (though approximate symmetries are common). The absence of global symmetries has powerful implications, for example requiring a complete spectrum of charged objects in quantum gravity [10–14]. Our goal in this paper is to argue that an apparently degenerate case of generalized symmetry, namely the case of $(-1)$-form U(1) global symmetry, is a useful concept that provides a unifying language for discussing many interesting dynamical phenomena in quantum field theory. Although this degenerate case has received relatively little attention in the literature, it has previously made an appearance in [15–19], and our discussion will build on ideas introduced therein.[1]

The case of $p$-form invertible global symmetries in $d$-dimensional quantum field theory, with $0 \leq p \leq d-2$, is well-established. A $p$-form global symmetry with group $G$ is associated with a family of topological operators $U(g, \Sigma)$, known as symmetry operators, labeled by a group element $g \in G$ and a closed (i.e., compact and without boundary) $(d-p-1)$-manifold $\Sigma$. These operators are topological, in the sense that correlation functions are invariant under deformations of $\Sigma$ provided that $\Sigma$ does not cross another operator insertion when deformed. The symmetry operator $U(g, \Sigma)$ acts on $p$-dimensional charged operators living on $p$-manifolds that are linked by $\Sigma$. Although the symmetry charge in general is defined on a $(d-p-1)$-manifold, in many cases it is localized to the $(p+1)$-dimensional worldvolumes of massive charged objects created by the charged $p$-dimensional operators. In the special case of a U(1) symmetry, the symmetry operators take the form

$$U(e^{i\alpha}, \Sigma) = \exp\left(i\alpha \int_\Sigma \star j_{p+1}\right), \tag{1}$$

where $j_{p+1}$ is a conserved $(p+1)$-form current. In other words, $j_{p+1}$ is co-closed:

$$\mathrm{d} \star j_{p+1} = 0. \tag{2}$$

---

[1]Other recent discussions of $(-1)$-form global symmetries, with less overlap with our current focus, appear in [20–22]. $(d-1)$-form global symmetry is another interesting special case; see [23].

The topological nature of the operator follows from this local conservation equation. (In some literature, the $(d-p-1)$-form operator $J_{d-p-1} = \star j_{p+1}$ is referred to as the conserved current, and it is closed rather than co-closed. Here we follow the classic convention in which an ordinary conserved current for a symmetry acting on local operators is a 1-form.) For a U(1) symmetry, the charge $Q = \int_\Sigma \star j_{p+1}$ is an integer, which is equivalent to saying that the right-hand side of (1) is invariant under $\alpha \mapsto \alpha + 2\pi$, a necessary condition for the operator to be a well-defined function of $e^{i\alpha} \in$ U(1).

Every symmetry is associated with topological operators, even if it is a traditional Noether symmetry that is conserved only on the equations of motion. The topological nature of the operator is a statement about correlation functions in the theory. However, in special cases the symmetry itself is topological in nature. For example, in Maxwell theory, the magnetic flux $\frac{1}{2\pi} \int_\Sigma F \in \mathbb{Z}$ is an integer topological invariant for any U(1) bundle and any closed 2-manifold $\Sigma$, without the need to use equations of motion. In such cases, the path integral decomposes into topological sectors, and the symmetry operator insertion is identical for all field configurations in a given sector. Whether a symmetry is topological in this sense can depend on the duality frame in which one works, so it is not a physical invariant. On the other hand, many ordinary symmetries are not topological in *any* duality frame. In this paper, we focus on symmetries that are topological in the strong sense, in the duality frame in which we define the theory.

A $p$-form U(1) global symmetry (without an 't Hooft anomaly) can be coupled to a background $(p+1)$-form gauge field (U(1) connection) $A_{p+1}$ by adding a coupling $A_{p+1} \wedge \star j_{p+1}$. We can gauge the symmetry by making $A_{p+1}$ dynamical, i.e., by summing over U(1) bundles with connection $A_{p+1}$ in the path integral (and generally including a kinetic term for $A_{p+1}$, which will typically be generated by loops). In this case, Maxwell's equation $\frac{1}{e^2} d\star F_{p+2} = \star j_{p+1}$ indicates that the would-be co-closed current has become co-exact, and as a result the symmetry operators become trivial.

## 1.1 Defining a $(-1)$-form U(1) global symmetry

The special case $p = -1$ of $p$-form global symmetry is somewhat degenerate, in a few senses. At first glance, one may reasonably be skeptical that it is a useful notion at all. A standard $p$-form global symmetry acts on $p$-dimensional charged operators. There is, apparently, no such thing as a $(-1)$-dimensional operator, so a $(-1)$-form symmetry would appear to have nothing to act on. On the other hand, a $p$-form symmetry is also associated to dynamical charged objects with a $(p+1)$-dimensional worldvolume, and we *do* have a notion of a dynamical object with a 0-dimensional worldvolume, namely, an instanton. (For example, it is common to speak interchangeably of D$(-1)$-branes or D-instantons in Type IIB string theory [24].) A related concern is that the symmetry operators $U(g, \Sigma)$ for a $(-1)$-form symmetry are associated with closed $d$-dimensional spacetime manifolds $\Sigma$; in other words, they are integrated over the entire spacetime. In this case, the question of whether the operator's correlation functions are topological is not obviously meaningful, because we cannot locally deform $\Sigma$ while keeping the spacetime background of our theory fixed.[2] Similarly, in the case of a continuous symmetry, a $(-1)$-form symmetry is associated with a conserved 0-form current, $d\star j_0 = 0$. However, this condition is trivial, because $\star j_0$ is a top form in the theory. Thus, *every* scalar operator in the theory defines, in some sense, a $(-1)$-form global symmetry, which threatens to render the concept vacuous. There may still be some merit to this concept, even in the extremely general case, where the absence of such $(-1)$-form global symmetries has been identified with the longstanding claim that there are no free parameters in quantum gravity [18].

---

[2]There may be useful perspectives in which the deformation occurs in configuration space, or in some type of auxiliary extra dimension. We will not make use of such perspectives in this paper.

In this paper, we focus on the case of $(-1)$-form U(1) global symmetries, which retain enough structure to be a useful concept [15–17, 19]. The U(1) case is associated with integer charges, $\int_\Sigma \star j_0 \in \mathbb{Z}$. Correspondingly, these theories can be coupled to a background axion field, i.e., a compact scalar $\theta(x) \cong \theta(x) + 2\pi$, which we think of as a 0-form U(1) gauge field. The gauge redundancies of $\theta$ are simply $\theta(x) \mapsto \theta(x) + 2\pi n$ for $n \in \mathbb{Z}$. These are the analogues of "large" or winding gauge transformations $A_p \mapsto A_p + 2\pi n \omega_p$ with $[\omega_p] \in H^p(M, \mathbb{Z})$ for a $p$-form gauge field; the axion has no analogue of the local gauge transformations $A_p \mapsto A_p + d\lambda_{p-1}$.[3]

We will follow the pragmatic approach of taking the possibility to couple a theory to a background axion field as our working *definition* of a $(-1)$-form U(1) global symmetry:

**Definition.** We say that a theory has a $(-1)$-*form* U(1) *global symmetry* when it contains an operator $j_0$ that can be consistently linearly coupled to a background field $\theta(x)$ taking values in a circle ($\theta \cong \theta + 2\pi$),

$$
e^{-S_E} \mapsto e^{-S_E} \exp\left( i \int_M \theta(x) \star j_0(x) \right). \tag{3}
$$

We refer to $j_0(x)$ as the $(-1)$-form U(1) symmetry current and (for the case of orientable spacetime manifolds $M$) we refer to $\int_M \star j_0(x) \in \mathbb{Z}$ as the $(-1)$-form symmetry charge.

The canonical example, and the case of greatest relevance to particle physics, is a $4d$ gauge theory with

$$
\star j_0 = \frac{1}{8\pi^2} \text{tr}(F \wedge F), \tag{4}
$$

for which the $(-1)$-form symmetry charge is the instanton number of a gauge field configuration. We will point out a number of other examples as we go along.

A few comments about our definition are in order. We require that the theory can be defined on arbitrary orientable spacetime manifolds $M$ for arbitrary $\theta(x)$ backgrounds, which in general are bundles over spacetime—i.e., they admit configurations in which $\theta(x)$ winds around a cycle.[4] In some cases, turning on other background fields can clash with turning on general $\theta$ backgrounds; in that case, we say that there is a mixed 't Hooft anomaly involving the $(-1)$-form symmetry, or an anomaly in the space of coupling constants, as discussed extensively in [15, 16].

We have referred to orientable spacetime manifolds because in a theory with an orientation-reversing spacetime symmetry like parity (by which we mean reflection of an odd number of spatial dimensions) or time reversal, there are additional subtleties. Such theories may be defined on non-orientable manifolds (see, e.g., [27–29] for recent discussions). In this case, the quantity that we can integrate over $M$ is a pseudoform or twisted form, i.e., one that transforms with an extra minus sign under parity. If $\star j_0$ is an ordinary form, then $\theta(x)$ must be a pseudoscalar in order for (3) to make sense. Our definition is valid in that case, but the charge $\int_M \star j_0(x)$ is not defined on arbitrary spacetime backgrounds, and in particular it does not make sense to couple the theory to a constant $\theta$-term on a general background. This is as expected: such a term violates parity.

The definition of a $(-1)$-form U(1) global symmetry that we have chosen is useful, because theories with this property have many features in common with theories with $p$-form U(1)

---

[3]See for instance [25].

[4]One might wonder if a weaker notion of $(-1)$-form symmetry is of interest, in which a theory need only admit a coupling to a *constant* background $\theta$ term. An interesting candidate is discussed in [26]: the $\mathbb{CP}^1$ sigma model in 3d has a topological invariant characterized by $\pi_3(\mathbb{CP}^1) \cong \mathbb{Z}$, which one might expect can be coupled to a constant $\theta$, but this topological invariant is not given by an integral of a local term. It turns out that the theory is only consistent with the choices $\theta = 0$ and $\theta = \pi$. We do not know any theory admitting a coupling to generic constant $\theta$ but not to a background axion.

global symmetries for higher $p$. For example, the symmetry can be gauged. In our context, we can do this by making the field $\theta(x)$ dynamical, summing over $\theta$ field configurations in the path integral. In some theories we can also gauge the symmetry by introducing massless chiral fermions [19]. In theses cases, as in ordinary electromagnetism, gauging the symmetry renders the current co-exact instead of merely co-closed. The central point of this paper is that the analogy also extends to the notion of spontaneous breaking of the global symmetry, and correspondingly, to higgsing (and dual confinement) when the symmetry is gauged.

For an ordinary U(1) global symmetry, it is possible to gauge a subgroup $\mathbb{Z}_k \subset U(1)$. This operation also extends to the $(-1)$-form case, where it corresponds to summing over only field configurations with topological charge a multiple of $k$ [17, 30–33].

## 1.2   Spontaneous breaking of a $(-1)$-form U(1) symmetry

The spontaneous breaking of $p$-form global symmetries for $p \geq 0$ has been extensively discussed in the literature [2, 34–36]. A standard diagnostic for breaking of an ordinary 0-form symmetry is that a charged operator obtains a vacuum expectation value. When the symmetry is continuous, we also find massless, propagating Nambu-Goldstone bosons that nonlinearly realize the symmetry. This picture extends to higher-form symmetries: for example, a Wilson loop generally has an expectation value that obeys a perimeter law or an area law. In the case of a perimeter law, a counterterm in the definition of the Wilson loop can cancel the perimeter dependence, leaving behind a constant expectation value even for arbitrarily large loops. This is the case of spontaneous breaking of a 1-form global symmetry, and the photon can be viewed as a massless Nambu-Goldstone mode. For a confining theory with an area law for the Wilson loop, on the other hand, the expectation value decays for large loops, and the 1-form global symmetry is considered to be unbroken.

Such diagnostics cannot be extended to the case of $(-1)$-form global symmetries, because there is no $(-1)$-dimensional charged operator that can obtain a vacuum expectation value. Similarly, there is no possibility of a propagating Nambu-Goldstone boson created by a $(-1)$-form field nonlinearly realizing the symmetry. Nonetheless, we will argue that there is a useful notion of spontaneous symmetry breaking for a $(-1)$-form U(1) symmetry, and even a sense in which there is an emergent Nambu-Goldstone *field* in the infrared (though not a propagating boson).

We propose that a useful diagnostic of spontaneous symmetry breaking for a $(-1)$-form global symmetry is that *the vacuum energy for the theory in Minkowski space depends on the value of a constant $\theta$ background*. In particular, one order parameter for such spontaneous symmetry breaking is the *topological susceptibility*, defined as

$$\mathcal{X} = -i \int d^d x \, \langle T\{j_0(x)j_0(0)\}\rangle_{\text{conn.}} = \frac{\partial}{\partial \theta}\langle j_0 \rangle = \frac{\partial^2}{\partial \theta^2} V(\theta), \tag{5}$$

where, conn. denotes the connected two-point function. One suggestive link between this expression and familiar cases of spontaneous symmetry breaking is that of the Kogut-Susskind pole [37, 38], which we briefly review here. Specifically, in many theories, the $(-1)$-form topological charge density $j_0$ is a total derivative of a (gauge-dependent) quantity $v_\mu(x)$,

$$j_0(x) = \partial^\mu v_\mu(x). \tag{6}$$

In this context, a nonzero value of $\mathcal{X}$ signals the existence of a pole in the two-point function of $v_\mu(x)$:

$$\mathcal{X} = \lim_{q \to 0} -i q^\mu q^\nu \int d^d x \, e^{iq \cdot x} \langle T\{v_\mu(x)v_\nu(0)\}\rangle_{\text{conn.}}, \tag{7}$$

which implies that

$$\lim_{q \to 0} \int d^d x \, e^{iq \cdot x} \langle T\{v_\mu(x)v_\nu(0)\}\rangle_{\text{conn.}} = i \frac{q_\mu q_\nu}{q^2} \frac{\mathcal{X}}{q^2} . \tag{8}$$

That is, the topological susceptibility is the residue of a pole at $q^2 = 0$ in a (gauge-dependent) two-point function. Because of the gauge dependence, this pole does *not* signal the existence of a propagating particle, but it does relate to important long-distance correlations in the theory [38].

An interesting perspective on the Kogut-Susskind pole is that it signals that the infrared theory has a description in terms of an emergent $(d-1)$-form gauge field [39–43]:

$$\star j_0 \to_{\text{IR}} dC_{d-1} . \tag{9}$$

A $(d-1)$-form gauge field has no propagating degrees of freedom, but there can be domain walls carrying a gauge charge under it, and such fields prove useful in various applications (see, e.g., [44–47]).

We would like to propose a reinterpretation of the emergent $(d-1)$-form gauge theory. In general, spontaneous breaking of a $p$-form U(1) global symmetry is associated with the emergence of a $p$-form Nambu-Goldstone gauge field in the IR. In the standard case of a 0-form symmetry, we think of this simply as a compact boson, but as we have argued, such bosons can also be thought of as 0-form gauge fields. A $p$-form gauge field has a complementary description in terms of a magnetic dual $(d-p-2)$-form gauge field, whose field strength is the Hodge dual of the original field strength:

$$\star d a_p \sim d b_{d-p-2} . \tag{10}$$

It is unclear what it would mean to seek a $(-1)$-form gauge field emerging in the IR description of a spontaneously broken $(-1)$-form global symmetry, but the magnetic dual makes perfect sense: it should be a $(d-1)$-form gauge field, precisely as in (9). Thus, we argue that the Kogut-Susskind pole can be thought of as signaling that the IR theory admits an emergent description in terms of a $(d-1)$-form Nambu-Goldstone gauge field. There is no Nambu-Goldstone *boson*, because there are no propagating degrees of freedom; nonetheless, the Nambu-Goldstone *field* can be useful.

One example of the utility of such a description arises when we gauge the $(-1)$-form global symmetry. When we gauge a spontaneously broken $p$-form global symmetry, the resulting theory is in the Higgs phase. We can summarize the phenomenon of higgsing and its magnetic dual, confinement, by saying that in this phase electrically charged worldvolumes *have* boundaries (they can end on a vacuum insertion) and magnetically charged worldvolumes *are* boundaries (they are confined by a higher-dimensional object). Many of the consequences of higgsing have analogues when we couple a dynamical axion field to a spontaneously broken $(-1)$-form global symmetry. To name a few:

- The gauge field acquires a mass. For the axion, this is apparent: there is a potential $V(\theta)$ and the axion mass is proportional to $\mathcal{X}$ at the minimum of the potential.

- Electric charges are screened. In the axion case, the electrically charged objects are instantons. We can think of the local operator $e^{i\theta(x)}$ as the analogue of a Wilson line: it inserts a static instanton configuration at a point. The effects of this insertion in correlation functions fall off at long distances, because the axion is massive.

- Magnetic charges are confined. In the axion case, these are vortices, codimension-two objects in spacetime around which the axion field winds. They are charged under the

gauge field $B_{d-2}$ dual to $\theta$. This field is eaten by the emergent $(d-1)$-form gauge field via a Stueckelberg structure of the form $|dB_{d-2} - C_{d-1}|^2$ [43]. Equivalently, axion vortices are the boundaries of domain walls, which carry charge under $C_{d-1}$.

We consider this set of parallels to be a strong argument that our definition of spontaneous breaking of a $(-1)$-form global symmetry is a useful one, allowing us to successfully apply intuition from more standard cases in a different context.

## 1.3 Application to the Strong CP problem

The language that we have introduced above provides a useful framework for thinking about the Strong CP problem. The Strong CP problem is the puzzle that the Standard Model admits a CP-violating term of the form $\frac{1}{8\pi^2}\bar{\theta}\int \text{tr}(G \wedge G)$,[5] but experiment finds that this term is extraordinarily small, $|\bar{\theta}| \lesssim 10^{-10}$ [48,49]. This cries out for some explanation in terms of symmetries or dynamics. Because the CKM phase in the quark mixing matrix has been measured to be an $O(1)$ number, the simplest answer that our universe respects CP is not a viable one. A number of solutions to this puzzle have been proposed over the years.

From our perspective, the Strong CP problem is closely related to the existence of a spontaneously broken $(-1)$-form U(1) global symmetry of the Standard Model, with charge the QCD instanton number. The symmetry itself allows us to turn on a $\bar{\theta}$ term (in the absence of an additional symmetry like CP, which would forbid a *constant* $\theta(x)$ background field on generic spacetimes). The spontaneous breaking of the symmetry allows $\bar{\theta}$ to affect physical observables like the neutron EDM. This suggests that a useful strategy for solving the Strong CP problem is to seek mechanisms for eliminating this global symmetry. A global symmetry can be eliminated by effects that explicitly break the symmetry, or by gauging. As already noted in [19], two different classic solutions to the Strong CP problem, the QCD axion and the massless up quark, can be understood as different ways of gauging the $(-1)$-form global symmetry. Explicitly breaking the symmetry is more challenging, since the underlying charge is topological. Nonetheless, there are physical mechanisms that can break such symmetries. We will discuss some of these mechanisms, and see that for the most part they do not offer a satisfactory resolution of the Strong CP problem. A final, classic set of solutions to the Strong CP problem rely on the spontaneous breaking of an orientation-reversing spacetime symmetry (parity or CP). These mechanisms, again, are linked to the fate of the $(-1)$-form symmetry, since the topological charge is not defined on the non-orientable spacetime backgrounds that are allowed in such theories.

## 1.4 Outline

The remainder of this text is structured as follows. In section 2 we present a discussion on how generic abelian gauge theories in the Coulomb phase can be understood as describing spontaneously broken higher form symmetries. We argue that this still holds in $2d$, where Maxwell theory realizes a spontaneously broken $(-1)$-form U(1) symmetry. We examine several deformations of this theory and propose universal features of spontaneously broken $(-1)$-form symmetries. In section 3 we argue that the instantonic symmetry of SU($N$) Yang Mills and QCD is spontaneously broken and link this fact with the Strong CP problem. In section 4 we explore solutions to the Strong CP problem from this point of view. We list some open questions and provide an outlook in section 5.

---

[5]The physical quantity $\bar{\theta}$ in fact is a linear combination of the coefficient of $\text{tr}(G \wedge G)$ and the phase of the determinant of the quark mass matrix; here we assume we have rephased the quarks to move the physical quantity entirely into the gluonic term.

# 2  Gauge theories as spontaneously broken phases

Standard lore holds that abelian gauge theories in the Coulomb phase describe spontaneously broken higher form symmetries. The lore further specifies that the Nambu-Goldstone bosons realizing the spontaneously broken symmetries nonlinearly are the photons themselves.[6] This section aims to show that this lore holds even in $2d$ theories, where the higher-form symmetry is a $(-1)$-form symmetry. To gain some intuition we review Maxwell's theory in $4d$ and $3d$, making our way to the two-dimensional world. Then we describe abelian gauge theories in $2d$ and give explicit realizations of the concepts introduced in Sec. 1. As we will describe, $2d$ gauge theories have instantons that are charged under a $(-1)$-form symmetry that is spontaneously broken.

## 2.1  $4d$ Maxwell theory

Free electromagnetism is a theory of a $U(1)$ gauge field $A$ with field strength $F = dA$ and the following action,

$$S = \int -\frac{1}{2e^2} F \wedge \star F \,. \tag{11}$$

The equation of motion for the gauge field is $d\star F = 0$. This equation signals the existence of a conserved 2-form current $J_e = \frac{1}{e^2} F$ that generates a 1-form $U(1)_e^{(1)}$ symmetry. The topological symmetry operator can be constructed by exponentiation of the current,

$$U_\alpha(\Sigma_2) = e^{\frac{i\alpha}{e^2} \int_{\Sigma_2} \star F} \,. \tag{12}$$

This symmetry operator counts the electric charge inside $\Sigma_2$ and acts by linking on non-dynamical probe electric charges dubbed Wilson lines. If massless electrically charged dynamical matter, such as the electron, is added to this theory, the electric charge of Wilson lines is screened and the $U(1)_e^{(1)}$ symmetry is explicitly broken. Provided that the gauge group is $U(1)$, electric charge is quantized and $\alpha \in [0, 2\pi)$, as befits a $U(1)_e^{(1)}$ symmetry. The gauge field obeys a topological constraint, the Bianchi identity $dF = 0$. As before, this equation signals the existence of a conserved 2-form magnetic current $J_m = \star F$ that generates a $U(1)_m^{(1)}$ symmetry. Exponentiation of the current yields the symmetry operators $\tilde{U}_\alpha(\Sigma_2)$ that measure the magnetic charge inside of $\Sigma_2$. If the gauge group is $U(1)$, $\oint \frac{F}{2\pi} \in \mathbb{Z}$, which is a topological invariant labeling gauge bundles by their monopole number. The topological nature of the magnetic symmetry makes explicitly breaking it a non-perturbative statement in the action in Equation (11). In other words, no modification of the Lagrangian, no matter how drastic it is, can explicitly break this symmetry as long as $A$ is a $U(1)$ gauge field.

A 1-form symmetry is generated by codimension 2 topological operators. There is no invariant way of defining an action of an operator of such dimensionality on local operators. This implies that a local operator can't transform under a 1-form symmetry. This is not true for local operators that are not gauge invariant, which do not correspond to physical observables. In fact the action of the electric 1-form symmetry can be encoded as a shift of the gauge field by a closed but not quantized 1-form $\Lambda_1$, $A \to A + \Lambda_1$, $\oint \Lambda_1 \in [0, 2\pi)$. Note that if $\Lambda_1$ was quantized this shift would correspond to a large gauge transformation. The gauge invariant operator transforming under the symmetry is the Wilson line, defined as $W_q(\gamma) = e^{iq \int_\gamma A}$ for some integer $q$ and its transformation rules follow from those of $A$. This is one of the nice features of gauge fields: they allow for the description of line operators in terms of local (but not gauge invariant) ones. A further important lesson follows from the transformation of $A$;

---

[6]This also holds for the compact scalar, which we understand as a gauge boson for a $(-1)$-form $U(1)$ symmetry. It nonlinearly realizes a spontaneously broken 0-form symmetry.

it realizes the 1-form symmetry non-linearly. This is a familiar property of Nambu-Goldstone bosons $\phi$ in phases with spontaneously broken U(1) 0-form symmetries. Given a symmetry transformation with compact parameter $c \in [0, 2\pi)$ the Nambu-Goldstone boson shifts as $\phi \to \phi + c$. The lesson that follows from this observation is that $A$ is the Nambu-Goldstone boson of a spontaneously broken $U(1)_e^{(1)}$ symmetry [2].

This heuristic observation can be made precise by computing the expectation value of an object charged under $U(1)_e^{(1)}$, a Wilson line. It obeys a perimeter law in the Coulomb phase, signaling the spontaneous breaking of the symmetry. The Goldstone theorem implies that, in a phase with a spontaneously broken U(1) 0-form symmetry, the conserved current creates Nambu-Goldstone bosons from the vacuum, which propagate a massless excitation. In the case at hand the conserved current creates a 1-form Nambu-Goldstone boson, the photon [2],

$$\langle 0|J_{e,\mu\nu}(x)|\lambda, p\rangle = \left(\lambda_\mu p_\nu - \lambda_\nu p_\mu\right) e^{ipx}. \tag{13}$$

The equation of motion and the Bianchi identity are exchanged under $\frac{1}{2\pi}F \leftrightarrow \frac{1}{e^2} \star F$. One can introduce a magnetic photon $\tilde{A}$ by defining a Hodge dual field strength $\frac{1}{2\pi}\tilde{F} = \frac{1}{e^2}F$ and a dual coupling $\tilde{e} = 2\pi e^{-1}$ and the action remains invariant. In the electric frame an 't Hooft line is defined as a boundary condition for the gauge field along a 1-dimensional locus. In the dual frame however it can be defined in terms of the dual gauge field $H_q(\gamma) = e^{iq\int_\gamma \tilde{A}}$. If one substitutes $U(1)_e^{(1)}$ with $U(1)_m^{(1)}$, $A$ with $\tilde{A}$ and the Wilson lines with 't Hooft lines all the discussion regarding the spontaneous breaking of the symmetry remains unchanged. It follows that both $U(1)_e^{(1)}$ and $U(1)_m^{(1)}$ are spontaneously broken in the Coulomb phase giving rise to a single Nambu-Goldstone boson in either the electric or magnetic frame.

It is interesting to note that the photon remains exactly massless even if both symmetries are explicitly broken at some scale by adding fundamental matter and dynamical monopoles. Indeed this is plausibly what happens in our universe. Since local operators may not carry 1-form charge, no relevant (or irrelevant) couplings can spoil the emergent 1-form symmetry, making it exact at low energies. This fact protects the masslessness of the photon. For related references see for instance [36, 50–56].

## 2.2 3d Maxwell theory

The symmetries of 3d electromagnetism follow a similar pattern to its 4d counterpart. There is a $U(1)_e^{(1)}$ 1-form symmetry under which Wilson lines are charged. In this case, however, the magnetic symmetry is 0-form $U(1)_m^{(0)}$. Magnetic charge is sourced by local operators called monopole operators which are defined by excising a point of spacetime and prescribing a boundary condition for $A$ sourcing magnetic flux. Due to the change in dimensionality the gauge field $A$ is dual to a compact scalar field $\sigma$. In terms of this field the monopole operator is defined as $M_p(x) = e^{ip\sigma(x)}$. In the 3d world a continuous 1-form symmetry can't spontaneously break and give rise to Nambu-Goldstone modes, a result which follows from a generalization of the Hohenberg-Mermin-Wagner-Coleman theorem [2, 34, 57, 58]. This implies that the electric 1-form symmetry can't be spontaneously broken in 3d. This result is made apparent by introducing dynamical monopoles which give rise to a confining force between electric particles [59]. The Wilson line then follows an area law, which in the large area limit vanishes, and the $U(1)_e^{(1)}$ symmetry remains unbroken. This means that the 3d photon can't be understood as the Nambu-Goldstone boson for the spontaneous breaking of the electric symmetry. This result follows from monopole proliferation, which in turn implies that the vacuum is magnetically charged and the magnetic 0-form symmetry is spontaneously broken. Furthermore, explicit computation in the magnetic frame shows that the matrix element between the magnetic current $J_{m,\mu} = (\star F)_\mu$ and the dual photon is,

$$\langle 0|J_{m,\mu}(x)|p\rangle = p_\mu e^{ipx}. \tag{14}$$

The lesson is that $3d$ electromagnetism can be understood as in the magnetic Coulomb phase and describes a spontaneously broken $U(1)^{(0)}$ symmetry.

As already mentioned, addition of magnetic monopoles leads to their proliferation and the onset of confinement of electric charges. A further effect of this proliferation is to give the photon a mass exponentially small in the monopole action $S_{\text{mon}}$. This is understood by noticing that once dynamical monopoles are included, the magnetic 0-form symmetry is only emergent at energies below $S_{\text{mon}}$. An emergent 0-form symmetry, unlike an emergent 1-form symmetry, is not enough to protect the masslessness of the photon. This was beautifully exemplified by Polyakov in [59]. He studied how the photon, when embedded in SU(2) through adjoint Higgsing gets a mass from vortices. The magnetic symmetry is absent in the SU(2) theory and, correspondingly, the U(1) photon is massive.

### 2.3   $2d$ Maxwell theory

We are now ready to tackle the wacky two dimensional world. In $2d$ the Maxwell theory admits a $\theta$-term, which we omitted in the $4d$ case. It will play a starring role in our discussion, so let us spell it out,

$$S = \int -\frac{1}{2e^2} F \wedge \star F + \frac{1}{2\pi} \int \theta F . \tag{15}$$

The equation of motion is unchanged, $d\star F = 0$, and gives rise to a conserved 2-form current $J_e$. The Bianchi identity is more subtle than in the higher dimensional counterparts. It still reads $dF = 0$ but it is a tautological equation, since every top form is closed. As in higher dimensions, the first Chern class of a U(1) gauge bundle in 2 dimensions is quantized $\oint F = 2\pi\mathbb{Z}$. This is what allows for the introduction of the $\theta$-term in the first place. We can use this fact to identify $2\pi F = \star j_0$ as a magnetic $(-1)$-form U(1) current and the $\theta$-term as the coupling to a background gauge field for it,[7] following our definition in Equation (3). In our terminology the symmetry of $2d$ Maxwell theory is then $U(1)_e^{(1)} \times U(1)_m^{(-1)}$.[8] Furthermore, in analogy with the higher dimensional counterparts, it is natural to expect the $(-1)$-form symmetry to be spontaneously broken. In the following we explore this possibility in detail. The field strength in $2d$ has a single component $F_{01}$, and the action can be rewritten.

$$S = \int d^2x \left[ \frac{1}{2e^2} F_{01}^2 + \frac{1}{2\pi} \theta F_{01} \right] . \tag{16}$$

It is useful to quantize the theory by choosing the space manifold to be a circle $S^1$ of radius R. By a suitable choice of gauge $A_0 = 0$ one can argue that only the zero mode survives and the theory can be rewritten in terms of an angular variable $\phi(t) = \int_0^{2\pi R} dx A_1(x, t)$. The angular nature follows from the large gauge transformations of $A$ winding along the circle, which become $\phi \to \phi + 2\pi$. In terms of $\phi$ the action becomes,

$$S = \int dt \left[ \frac{1}{4\pi e^2 R} \dot{\phi}^2 + \frac{\theta}{2\pi} \dot{\phi} \right] . \tag{17}$$

This is just the action for a particle in a circle in the presence of a magnetic field. The system can be quantized and has energy eigenstates $\psi_l = e^{il\phi}$ with energy,

$$E_l = \pi e^2 R \left( l - \frac{\theta}{2\pi} \right)^2 . \tag{18}$$

---

[7]Given that $\theta$ is a background gauge field, the meaning of the transformations $\theta \to \theta + 2\pi$ is clear, they are just the large gauge transformations of the background gauge field. These large gauge transformations shift the scalar background gauge field by a closed but not exact 0-form: a constant. In the $(-1)$-form symmetry case this is all there is, since small gauge transformations, described by shifts by an exact 0-form, are trivially zero.

[8]A SymTFT realization of this symmetry can be found in appendix C.

The ground state is $\psi_0$, which we denote $|0\rangle = |\psi_0\rangle$. The system in state $\psi_l$ is characterized by a constant electric field,

$$F_{01} = e^2 \left( l - \frac{\theta}{2\pi} \right). \tag{19}$$

The ground state is the state with lowest electric field,

$$\langle F_{01} \rangle_0 = -\frac{e^2 \theta}{2\pi}. \tag{20}$$

This result is hardly surprising since the classical equations of motion for $A_0, A_1$ in eq. (16) are $\partial_0 F^{01} = \partial_1 F^{01}$, whose only solution is a constant electric field. This also agrees with the photon not propagating any degree of freedom in $2d$. In free Maxwell theory there are no electric particles but one can consider adding heavy particles, or Wilson lines, to probe the theory. On the circle we must add at least two, of opposite charge and separated by a distance $L$. Regardless of the chosen state, the electric field between them will increase by one unit, giving rise to an energy that grows linearly with $L$. This shows that, even if the photon does not propagate any degree of freedom there is a long range force that confines probe charges classically.

Classic confinement implies that large Wilson loops obey an area law and the electric 1-form symmetry is not spontaneously broken. A similar check is not available for the magnetic $(-1)$-form symmetry. We would need a charged operator that is analogous to the 't Hooft loop in $4d$ or the monopole operator in $3d$ but such a thing does not seem to exist. As discussed in Sec. 1, we propose instead that the spontaneous breaking of the $(-1)$-form symmetry is diagnosed by an explicit dependence of the vacuum energy $V(\theta)$ on the value of the background $\theta$. The leading measure of such dependence is the topological vacuum susceptibility $\mathcal{X} = \frac{\partial^2}{\partial \theta^2} V(\theta)$. Given the topological density $\star F$, the topological susceptibility can be rewritten as,

$$\mathcal{X} = -i \frac{1}{4\pi^2} \int d^2 x \langle T(\star F(x) \star F(0)) \rangle_{\text{conn.}} = \frac{1}{2\pi} \frac{\partial}{\partial \theta} \langle \star F \rangle = \frac{e^2}{4\pi^2}, \tag{21}$$

which is nonzero in the present case, signaling spontaneous breaking of the $(-1)$-form symmetry. So far we have linked the spontaneous breaking of the $(-1)$-form symmetry with a physical dependence on its gauge background. A further motivation for this definition is the relation between a non-vanishing $\mathcal{X}$ and the appearance of a double pole at zero momentum in the 2-point function of the photon. By considering the gauge-dependent two point function $\langle A_\mu(x) A_\nu(y) \rangle$, one can show that it is written in terms of a propagator $G(q^2)$ that satisfies,

$$\lim_{q^2 \to 0} q^2 G(q^2) \sim \mathcal{X}. \tag{22}$$

Although $G(q^2)$ is gauge dependent, this limit matches the manifestly gauge invariant quantity (21) and so the pole at $q^2 = 0$ is independent of the gauge [38]. While in higher dimensions the Kogut-Susskind pole is somewhat mysterious, there is no mystery in the abelian two dimensional case where the role of the non-propagating photon field is well understood. In particular, it is responsible for the long-range force that confines probe particles. In this sense, the gauge field $A_\mu$ mediates a long range force thanks to a pole in its "propagator," in complete analogy with Maxwell theory in higher dimensions. We conclude that the non-vanishing of the topological susceptibility signals the spontaneous breaking for the $(-1)$-form symmetry giving rise to a Nambu-Goldstone *field* that creates a long range force, even if it does not propagate.

After explicitly establishing the connection between free $2d$ Maxwell theory and the $(-1)$-form symmetries introduced in sec. 1, in the following we explore the fate of these universal features in theories with fermions, both massless and massive. We have also considered two other $2d$ models displaying interesting low energy dynamics that can be understood in terms of

the magnetic $(-1)$-form symmetry but, for deference to the exhausted reader, those discussions are left to the appendices. In appendix A we review the $2d$ Abelian-Higgs model while in appendix B we review the $\text{CP}^{N-1}$ model.

## 2.4 The Schwinger model

If we couple the $2d$ U(1) gauge theory with a charge 1 massless Dirac fermion we obtain the Schwinger model. The action is,

$$S = \int -\frac{1}{2e^2} F \wedge \star F + \frac{1}{2\pi} \theta F + i\bar{\psi}\slashed{D}\psi \,. \tag{23}$$

Explicit computation of the equation of motion shows that the electric current is no longer conserved. There is a would-be chiral U(1) symmetry that is ABJ anomalous. Finally, despite the $\theta$-term, we will argue below that the would-be $\text{U}(1)_m^{(-1)}$ symmetry of this theory is gauged, not global.

The gauge coupling is dimensionful in two dimensions and the theory is strongly coupled in the IR. Nonetheless, it is simple enough for Schwinger to be able to solve it explicitly using the operator formalism [60]. If you are not Schwinger, a simpler approach was pioneered by Coleman [61] taking advantage of $2d$ bosonization. This duality states that a strongly coupled fermion can be exchanged with a weakly coupled boson, provided that a dictionary is used.[9] The fermion theory confines classically at low energies but it is equivalently described by a theory of a free compact scalar which only couples to the gauge field through a topological term. In the present case the bosonized version of the theory takes the following form,[10]

$$S' = \int -\frac{1}{2e^2} F \wedge \star F + \frac{1}{2\pi}(\theta + \phi)F + \frac{1}{8\pi}(d\phi)^2 \,, \tag{24}$$

with $\phi$ a scalar of period $2\pi$. One could naively think that there is still a ground state electric field given by,

$$F_{01} = -\frac{e^2}{2\pi}(\theta + \phi)\,. \tag{25}$$

However, we can redefine $\phi \to \phi - \theta$ to absorb $\theta$, signaling that $\theta$ is unphysical. In fact, this was already apparent in the original formulation of the theory, which has an ABJ anomaly that allows $\theta$ to be absorbed in a chiral rotation. The bottom line is that the electric field in the ground state, which was proportional to $\theta$ in the free Maxwell case, can now relax to a vanishing value.

$$\langle 0|(\star F)|0\rangle = 0\,. \tag{26}$$

In more detail, in $2d$ the ABJ anomaly is computed by the vacuum polarization diagram. The vacuum polarizes and the electric field is screened. We highlight that this screening is not mediated by Schwinger pair production since the electric field to be screened is fractional in units of the charges of the massless fermions.[11] Consequently, the topological susceptibility $\mathcal{X}$

---

[9]An interesting comment, made to us by a SciPost referee is the following. In general the bosonization dictionary involves gauging the fermion parity in the fermionic theory. In the present case the fermion parity is already gauged in the Schwinger model and the equivalence with the bosonic theory is faithful.

[10]The dictionary fixes the periodicity of the canonically normalized compact scalar. A free Dirac fermion bosonizes to a canonically normalized compact scalar with period $\sqrt{\pi}$ [62], hence the factor of $1/(8\pi)$ in the kinetic term in (23).

[11]One can also argue for this by noticing that the constant electric field becomes $F_{01} = \frac{-e^2}{2\pi}\phi$, which gives a non-zero energy. Another way to argue for the field relaxing to zero is by integrating out $F = dA$ from 24. One finds a quadratic potential for $\phi$, which naturally relaxes to zero setting $F_{01} = 0$. For related discussions see for instance [63,64].

vanishes and the Schwinger model does not spontaneously break the $(-1)$-form symmetry. By virtue of eq. (22) the vanishing of the topological susceptibility implies that the pole disappears from the gauge 2-point function, signaling that the Nambu-Goldstone *field* has been lifted and we are no longer in the Coulomb phase. Hence we also expect the long-range force between probe particles to vanish.

Indeed the polarized vacuum can also screen the electric field sourced by probe particles and the long-range force is well known to be absent in the Schwinger model [60, 65]. We see that all our expectations regarding a theory that spontaneously breaks a $(-1)$-form symmetry are negated in this model.

We finish our discussion by noting that the Schwinger model is equivalent, through the bosonization dictionary, to a theory where the $(-1)$-form symmetry has been gauged. In eq. (24) the $(-1)$-form symmetry current $J_0 = \star F$ has been coupled to a dynamical gauge field $\phi$ (a compact scalar), which is the canonical way of gauging symmetries associated to conserved currents. From this point of view, it is very natural that the Schwinger model cannot possibly realize a spontaneously broken $(-1)$-form symmetry, since it has been gauged! This observation will be useful when we leverage the knowledge gained from this toy model to understand QCD and the Strong CP problem.

## 2.5 The massive Schwinger model

A further twist can be made by adding a mass to the Dirac fermion in the Schwinger model. In the massless Schwinger model the vanishing of the topological susceptibility was intimately tied with the ABJ anomaly, which is absent in this case due to the fermions having a mass. For this reason, we expect this model to realize a spontaneously broken $(-1)$-form global symmetry. If the fermion is massive enough $m^2 \gg e^2$, we recover free Maxwell theory in the IR and our expectation is trivially satisfied. A more interesting question is what happens in the opposite case of $m^2 \ll e^2$. Consider the following action of a massive Dirac fermion coupled to a U(1) gauge field,

$$S = \int -\frac{1}{2e^2} F \wedge \star F + \frac{1}{2\pi} \theta F + i \bar{\psi} \slashed{D} \psi - im\bar{\psi}\psi \,. \tag{27}$$

In the $m^2 \ll e^2$ regime the theory is strongly coupled in the IR. Luckily, Coleman taught us how to solve it using the bosonization dictionary. The bosonized theory is [61, 66],

$$S' = \int -\frac{1}{2} F \wedge \star F + \frac{1}{2\pi}(\theta + \phi)F + \frac{1}{8\pi}(d\phi)^2 + \frac{m}{\pi\epsilon}\cos\phi \,, \tag{28}$$

where $\epsilon$ is a UV regulator [62, 67, 68]. The only difference with the bosonized action of the massless Schwinger model is the last term in Equation (28) which is absent in Equation (24). This term obstructs the absorption of $\theta$ by a $\phi$ field redefinition, so we expect $\theta$ to be physical and to give rise to a vacuum electric field. As discussed in [61, 66], this is precisely what happens. There is a non-zero electric field in the vacuum, which can't be screened in this case,

$$\langle 0|(\star F)|0\rangle = \frac{e^2}{2\pi}(\theta + \phi)\,. \tag{29}$$

From Equation (21) it follows that there is a non-zero topological susceptibility and, consequently a zero momentum pole in the 2-point function of the gauge field. Furthermore, this theory displays a long-range force between probe particles. The bottom line is by now clear, the magnetic $(-1)$-form symmetry, which was gauged in the massless case, is now ungauged and spontaneously broken, giving rise to a Nambu-Goldstone *field* and a long-range force.

A difference with the pure Maxwell case is that the long range force between two particles vanishes if the difference of their charges is a multiple of 2. The reason is that a Schwinger pair of massive fermions may nucleate, screening the electric field created by the particles. We learn that the long range characteristic of spontaneously broken $(-1)$-form symmetries may be dynamically screened but will be present for improperly quantized probes. A similar phenomenon happens with the vacuum electric field, giving a physical explanation for the periodicity of $\theta$ in this model.

An explicit, numerical computation of the topological susceptibility was carried out in the recent work [64] by using a tensor network approach in the lattice, where it was indeed found to be non-vanishing.

# 3 A different look at the Strong CP problem

While 2 dimensions are fun, they are somewhat detached from the high energy physics of our universe. In this section we present two 4$d$ gauge theories whose low energy dynamics are governed by a spontaneously broken $(-1)$-form symmetry. Namely, SU($N$) Yang-Mills theory and QCD. We will see how the low energy dynamics in these theories have similarities with the physics of 2$d$ Maxwell theory, particularly in the large $N$ limit of Yang-Mills. In the last part of this section we use this insight to reformulate the Strong CP problem as a consequence of the spontaneous breaking of a $(-1)$-form symmetry.

## 3.1 Low energy effective theory in 4d Yang-Mills theory and QCD

For concreteness, let us consider SU($N$) Yang-Mills theory with no light matter, which serves us as a toy model for QCD. The action is,

$$S = \int \text{tr}\left(-\frac{1}{g^2} F \wedge \star F + \frac{\theta}{8\pi^2} F \wedge F\right). \tag{30}$$

An important property of SU($N$) YM theory, is that the $\theta$-term leads to physical effects. Note that this property lies at the core of the Strong CP problem. Such a physical dependence on $\theta$ is probed by the topological susceptibility of the vacuum $\mathcal{X}$ which can be defined as [69],

$$\mathcal{X} \equiv \left(\frac{d^2 E}{d\theta^2}\right) = \lim_{q \to 0} -i \left(\frac{1}{16\pi^2}\right)^2 \int d^4 x \, e^{iqx} \langle 0|T(\text{tr}(F_{\mu\nu}\tilde{F}^{\mu\nu}(x))\text{tr}(F_{\rho\sigma}\tilde{F}^{\rho\sigma}(0)))|0\rangle. \tag{31}$$

The low energy dynamics of 4$d$ Yang-Mills theory is strongly coupled and notoriously difficult to study. Nonetheless, the non-vanishing of the topological susceptibility for generic $\theta$ gives us important hints about the vacuum structure. As noticed by Lüscher in [38], the 2-point function above can be recast in terms of a 2-point function of the Chern-Simons 1-form current $K_1$, which is the Hodge dual of the Chern Simons 3-form $K_1 = \star C_3$, at vanishing momentum. Using the fact that $\partial^\mu K_\mu = \frac{1}{16\pi^2}\text{tr}(F_{\mu\nu}\tilde{F}^{\mu\nu})$ one can rewrite eq. (31) as,

$$\mathcal{X} = \lim_{q \to 0} -iq^\mu q^\nu \int e^{iqx} \langle 0|TK_\mu(x)K_\nu(0)|0\rangle d^4 x. \tag{32}$$

As discussed in the introduction, the non-vanishing of the topological susceptibility implies that the two point function has a pole at $q^2 = 0$. Given that the 2-point function in question is not gauge invariant, one may be wary that this pole may be unphysical. Luscher argued that the pole remains in any gauge and, as we will explain, its physical implications are profound. The existence of this pole at zero momentum signals the appearance of a massless mode for $C_3$

in the IR. Since we believe that Yang-Mills theory is otherwise gapped, it is natural to expect the vacuum structure of SU($N$) Yang-Mills theory to be captured by an effective theory of a massless 3-form gauge field $C_3$. In fact, this observation should hold for QCD as well, since $\mathcal{X}$ is also nonzero in that case. For related discussions in Yang-Mills and QCD see [40, 70–72].

The precise form of the Lagrangian describing this effective theory will depend on strongly coupled dynamics and is, in general, not available to us. In the large N limit matters are simpler, as discussed in [40, 41, 70, 71]. At small momenta only terms with less than two derivatives will be relevant. One can assume that a kinetic term is generated and all other 2-derivative terms are in fact suppressed in the large N limit. In this limit the effective theory takes the following form,

$$\mathcal{L} = -\frac{1}{2\mathcal{X}} F_4 \wedge \star F_4 + \frac{1}{2\pi} \theta F_4 \,. \tag{33}$$

Where $F_4 = \mathrm{d}C_3$ is an abelian 4-form field strength that should not be confused with $F$, the 2-form non-abelian field strength. This action describes a 3-form gauge field in $4d$ with a topological coupling or $\theta$-term. This theory is reminiscent to electromagnetism in $2d$, see 2.3. In fact, the physics of both theories is very similar, as explored in, for instance [73]. Like its $2d$ counterpart, a 3-form gauge field in $4d$ does not propagate and the different vacua are characterised by a constant electric field,

$$\langle \star F_4 \rangle_l = (\theta + 2\pi l)\mathcal{X} \,. \tag{34}$$

The energy density of these vacua is

$$E_l(\theta) = \frac{1}{2}(\theta + 2\pi l)^2 \mathcal{X} \,. \tag{35}$$

The true vacuum, or ground state, is selected by minimizing the expression above. One then finds a non-zero electric field in the ground state,

$$\langle \star F_4 \rangle_0 = \mathcal{X}\theta \,. \tag{36}$$

Reassuringly these results match the expectations that one infers from holography [74]. Away from the large $N$ limit, the precise form of the action for such a field is unknown.[12] In general the Lagrangian will take the following form,

$$\mathcal{L} = -\frac{1}{2}|F_4|^2 + \frac{1}{2\pi} \theta F_4 + \mathbf{K}(F_4) \,, \tag{37}$$

where $\mathbf{K}(F_4)$ denotes higher order contributions with $F_4$. For instance, in QCD $\theta$ and $\mathbf{K}(F_4)$ will depend explicitly on the quark masses. Regardless of the precise form of the Lagrangian the equations of motion still admit a constant solution for $\star F_4$ such that the physical picture remains unchanged.

## 3.2 SSB of the ($-1$)-form U(1) symmetry in 4d Yang-Mills theory and QCD

An important property of any theory with a Lagrangian of the form 37 is that it has a magnetic ($-1$)-form U(1) symmetry. The existence of this symmetry follows from the Bianchi identity of the $C_3$ gauge field and the quantization of $\oint F_4$. From the Biachi identity we identify the conserved current as,

$$j_0 = \star F_4 \,, \tag{38}$$

---

[12]Its form may be determined in some approximations such as the dilute instanton gas; see [43].

which we can couple to a background gauge field $\theta$ which is periodic. This symmetry in the IR effective theory is already present in the UV of both SU($N$) Yang-Mills and QCD: it is a Chern-Weil symmetry with conserved current [19],

$$\star j_0 = \frac{1}{8\pi^2}\text{tr}(F \wedge F), \tag{39}$$

where now $F$ is the non-abelian field strength. This symmetry is sometimes called the instantonic symmetry, as it measures the instanton number. The non-vanishing of the topological susceptibility in SU($N$) YM and QCD implies that the physics depends non-trivially on the value of the background field $\theta$. Following our discussion in section 2.3 we take this dependence to signal the spontaneous breaking of the $(-1)$-form U(1) symmetry. Finally, we identify the 3-form gauge field $C_3$ as the Nambu-Goldstone *field* of the spontaneously broken $(-1)$-form symmetry.

A further way of arguing that the $(-1)$-form symmetry is spontaneously broken is by considering its gauging. It can be explicitly gauged by introducing a kinetic term for the gauge field $\theta(x)$ and summing over it in the path integral. This is equivalent to coupling the Chern-Weil current to an axion. Importantly, the axion has a non-trivial potential arising from the $\theta$ dependence of the vacuum energy density, i.e., the non-vanishing $\mathcal{X}$. This potential endows the axion with a non-zero mass, signaling that the *gauged* $(-1)$-form symmetry is spontaneously broken giving rise to a Higgs mechanism. Furthermore, an electric Higgs phase is dual to magnetic confinement. This can be explicitly checked in this case by replacing $\theta(x)$ by its Hodge dual 2-form gauge field $d\theta \sim \star dB_2$. The 3-form field $C_3$ is no longer massless as it picks up a mass from a Stueckelberg-like mass term of the form,

$$|dB_2 - C_3|^2, \tag{40}$$

which implies that the would-be Nambu-Goldstone field $C_3$ is eaten by the gauge field $B_2$. A similar Stueckelberg-like mass is obtained in, e.g., the dual description of gauging the magnetic 1-form U(1) symmetry. From 40 it follows that, to preserve gauge invariance, axion strings must be attached to $C_3$ domain walls. These domain walls have a finite tension, giving rise to a confining force between strings. We conclude that axionic strings are confined, in agreement with the gauge $(-1)$-form symmetry being spontaneously broken and Higgsed. Note that, were $\mathcal{X}$ to vanish, the effective field theory would not have a massless 3-form gauge field which we have associated with the spontaneous breaking of the $(-1)$-form U(1) symmetry. Furthermore, in the gauged $(-1)$-form symmetry theory, the axion would remain massless and the $(-1)$-form symmetry unhiggsed. These two facts, together with similar considerations in section 2 lead us to propose $\mathcal{X}$ as an order parameter for the spontaneous breaking of the $(-1)$-form symmetry.

> **Spontaneous Breaking of a $(-1)$-form U(1) symmetry.**
>
> A $(-1)$-form U(1) symmetry with background gauge field $\theta$ is spontaneously broken if the vacuum energy in Minkowski space $V$ depends on $\theta$. An order parameter for such spontaneous breaking is the topological susceptibility:
>
> $$\mathcal{X} = \frac{\partial^2}{\partial\theta^2}V(\theta). \tag{41}$$

## 3.3 Reformulation of the Strong CP problem

An important consequence of the non-vanishing of $\mathcal{X}$ in QCD is that physical observables can depend on $\bar{\theta} = \theta + \text{Arg}(\det\mathcal{M})$, where $\mathcal{M}$ is the quark mass matrix. An example of such

observable is the neutron electric dipole moment (nEDM). Experimental measurements of the nEDM place stringent constraints,

$$|\bar{\theta}| \lesssim 10^{-10}\,. \tag{42}$$

In the following we will refer to $\bar{\theta}$ as $\theta$. That is, we take $\theta$ to be the physical parameter. Given that $\theta$ is a free angular parameter of the quantum theory, its experimental value is unnaturally small, giving rise to the Strong CP problem. We have argued that the non-vanishing of $\mathcal{X}$ and, more generally, a partition function[13] that depends on $\theta$ signals the spontaneous breaking of the $(-1)$-form U(1) Chern-Weil symmetry of QCD. More broadly, a theory with a spontaneously broken $(-1)$-form symmetry has a physical dependence on a circle valued background field $\theta$. If $\theta$ is measured to be too small, a naturalness problem arises. The Strong CP problem is the QCD avatar of this naturalness problem. We can now extract a necessary condition for the QCD Strong CP problem to arise.

> **A necessary condition for the Strong CP problem in QCD.**
>
> A necessary condition for Quantum Chromodynamics to have a Strong CP problem is that the global $(-1)$-form U(1) symmetry is spontaneously broken.

In the next section we will use this necessary condition for the Strong CP problem to arise in QCD to provide a new perspective on the problem and its solutions.

## 4 Solutions to the Strong CP problem and its analogues

We have argued that the Strong CP problem is intimately tied with a spontaneously broken $(-1)$-form U(1) symmetry that arises in the Standard Model. If the physics associated with this spontaneous breaking is prevented in some way, the Strong CP problem should be solved. This problem has direct analogues in various other theories with spontaneously broken, global $(-1)$-form symmetries, some of which are easier to analyze because they are low-dimensional, as we have discussed above. In this section, we discuss various solutions to the Strong CP problem from this perspective.

### 4.1 Solving the problem by gauging with an axion

The classic Peccei-Quinn-Weinberg-Wilczek solution to the Strong CP problem [75–78] may be thought of as *gauging* the $(-1)$-form global U(1) symmetry with a dynamical axion field $\theta(x)$. The existence of a $(-1)$-form global U(1) symmetry means that our theory can be consistently coupled to a background axion field $\theta(x)$; we now simply sum over all such possible backgrounds in the path integral.

In the analogue problem in $2d$ Maxwell theory, we have already introduced the relevant action in (24), where the field $\phi$ plays the role of the dynamical axion. Such a coupling explicitly removes the physical dependence on $\theta$ by polarizing the vacuum and screening the constant electric field. In this case, the original 0-form U(1) gauge symmetry is Higgsed. One can see this explicitly by dualizing $\phi$ to $\tilde{\phi}$. The resulting kinetic term is $\sim |\mathrm{d}\tilde{\phi} - A|^2$, which shows that $A$ is made massive by a Stueckelberg mechanism. However, the $(-1)$-form U(1) gauge symmetry is *also* higgsed, eliminating the Kogut-Susskind pole. We can see this by dualizing the field strength $F = \mathrm{d}A$ to a 0-form integer field strength $n$, which acquires a Stueckelberg-type "kinetic term" $|n - \phi|^2$ that can also be interpreted as a potential that makes the gauge field $\phi$ for the $(-1)$-form symmetry massive. In general, we expect higgsing of a $(-1)$-form gauge symmetry to correspond to confinement of axion vortices by domain

---

[13]That is a generating "functional" with $\theta$-term as an external source term.

walls. In the $(1+1)d$ case, the domain walls are simply particles charged under $A$, while the operator $e^{i\tilde{\phi}}$ inserts a static vortex at a point in spacetime. Because $\tilde{\phi}$ shifts under $A$ gauge transformations, such a vortex *must* have an attached domain wall. This is the expected dual confinement phenomenon. Higher-dimensional analogues of this have been extensively discussed in the literature on inflation [46, 47].

For the Strong CP problem in QCD, the relevant coupling takes the following form:

$$S \supset \frac{1}{8\pi^2} \int \theta(x) \text{tr}(G \wedge G). \tag{43}$$

The Vafa-Witten theorem [79] ensures that the axion potential generated by QCD dynamics sets the effective low-energy $\theta$ angle to zero. As in the 2$d$ case, this term describes the coupling of the $(-1)$-form symmetry current $j_0 = \frac{1}{8\pi^2} \star \text{tr}(G \wedge G)$ to a dynamical gauge field, the axion. The net effect is that the $(-1)$-form symmetry is gauged. The axion equation of motion then shows that the instanton number current becomes co-exact,

$$f^2 \text{d}\star\text{d}\theta = \frac{1}{8\pi^2} \text{tr}(G \wedge G). \tag{44}$$

This exactness condition is equivalent to gauging: an exact current integrates to zero on any closed manifold, implying that there are no charged operators that may link with the symmetry operators $e^{i\oint \star j}$. There are thus no objects charged under a symmetry generated by an exact current. This is the chief property of a gauge symmetry in gauge theory. As discussed in §3.2, the gauged $(-1)$-form symmetry is in a higgsed phase, which is reflected in the confinement of magnetically charged objects (axionic strings) by axion domain walls.

It is worth noting that the Strong CP problem may not be solved if the axion potential has additional contributions beyond the QCD one. In this case one says that the axion suffers a quality problem. In our language the failure boils down to the fact that the $(-1)$-form U(1) global symmetry is not automatically gauged anymore. Indeed eq. (44) ceases to hold generically. For a discussion in greater detail see [19].

## 4.2 Solving the problem by gauging with massless fermions

A second canonical solution to the Strong CP problem is to postulate a chiral massless fermion. In the case of 2$d$ Maxwell theory the resulting theory is the Schwinger model, whose action we wrote in (23). Such a massless chiral fermion comes with an ABJ anomaly for the chiral symmetry that allows the $\theta$ angle to be rotated away, making it an unphysical parameter. This effect is particularly explicit in the 2$d$ case thanks to the 2$d$ bosonization by which the Schwinger model is equivalent to the bosonic theory in eq. (24), where the $\theta$ is absorbed by a redefinition of the compact scalar field. It is now clear what is happening in terms of the $(-1)$-form symmetry. The $(-1)$-form symmetry has been gauged, making its spontaneous breaking innocuous. It follows that the ground state electric field is screened by a polarized vacuum and the Strong CP problem is avoided. In 2 dimensions it is clear that these two solutions to the Strong CP problem are really the same. Furthermore, the lesson that adding massless fermions gauges $(-1)$-form symmetries holds more generally. For instance, in Yang-Mills, adding a massless fermion produces an ABJ anomaly for the chiral current $J_c$,

$$\text{d}\star J_c = \frac{1}{8\pi^2} \text{tr}(F \wedge F). \tag{45}$$

This equation implies that $\star j_0 = \frac{1}{8\pi^2} F \wedge F$ is (globally) exact, and hence (as explained in [19]) the $(-1)$-form symmetry is gauged. As in Sec. 4.1, this gauged $(-1)$-form symmetry is in a Higgs phase. In this case, although there is no elementary axion field, there is still a magnetic confinement phenomenon. The confined vortices are the boundaries of $\eta'$ domain walls, which have chiral excitations carrying baryon number, as described in [80].

## 4.3 Solving the problem with non-compact symmetries

An alternative solution to the CP problem was proposed in [81]. As discussed there, if one considers a 2$d$ abelian gauge theory with gauge group $\mathbb{R}$ instead of U(1), the analogue of the Strong CP problem is immediately solved. The reason for this to work is most easily understood in terms of the $(-1)$-form symmetry, which we recall, is the magnetic symmetry of the U(1) gauge theory in 2$d$. As is well known, for the gauge group $\mathbb{R}$ the would-be magnetic symmetry operators act trivially.[14] In more detail, there is a topological constraint that $\int_M F = 0$ for any closed 2-manifold $M$, so a $\int_M \theta F$ term for *constant* $\theta$ does not affect the physics. In particular, the vacuum energy is independent of constant $\theta$ and so we would not say that the theory spontaneously breaks a $(-1)$-form symmetry. Physically, a background electric field on a non-compact space can be screened by combinations of particles with mutually irrational electric charges. An analogous 4$d$ setup to this 2$d$ theory requires modifying the instanton sum by coupling to a topological theory (TQFT) with a non-compact 3-form gauge field [33]. As in the 2$d$ theory however, a dynamical mechanism, i.e., adding mutually irrationally charged domain walls, is needed to relax $\theta$ and fully solve the Strong CP problem.

For the Strong CP problem of the Standard Model, [81] also proposed a related mechanism, relying on a *non-compact axion* field $a(x)$, which has couplings

$$\int \frac{1}{8\pi^2} \left[ \xi_H a(x) \mathrm{tr}(G_H \wedge G_H) + \xi a(x) \mathrm{tr}(G \wedge G) \right] . \tag{46}$$

Here $G(x)$ is the usual Standard Model gluon field strength, while $G_H$ is the field strength of a hidden Yang-Mills group that confines at a much higher scale. This confinement generates a potential with a set of minima for $a(x)$. If $\xi_H$ and $\xi$ are mutually irrational, then the infinite set of minima of the $G_H$-generated potential allows the effective theta term of QCD to scan over a dense discretuum of values, some of which will be very small. One then must invoke a cosmological argument for why we find ourselves in a universe with such a small value. In our language, this model has gauged a $(-1)$-form $\mathbb{R}$ global symmetry, which is an irrational combination of two $(-1)$-form U(1) global symmetries. This is only possible with an axion field that is non-compact.

The common feature of the models of [33, 81] is the introduction of non-compact gauge fields (either an ordinary gauge field, or an axion, or a three-form field). This can enable novel solutions of CP problems in quantum field theory, but we expect that such models do not have consistent UV completions in quantum gravity (see, e.g., [11]).

## 4.4 Failing to solve the problem with explicit breaking

As we have discussed, standard solutions to the Strong CP problem rely on gauging the $(-1)$-form global symmetry. One might wonder if, instead, we could simply break the symmetry explicitly. In the following we discuss a couple of strategies that implement this idea but that ultimately fail to solve the problem.

---

[14]Equivalently one may say that an $\mathbb{R}$ gauge theory is obtained from the U(1) theory by performing a topological gauging of the U(1) magnetic symmetry. This gauging is enforced by coupling the magnetic current to a non-dynamical (i.e. with no kinetic term) U(1) gauge field with only flat connections and summing over it. In the present case this auxiliary field is a flat compact scalar. From this point of view, the Strong CP problem is avoided also in this case by gauging the $(-1)$-form symmetry. A SymTFT discussion of this model and the topological gauging can be found in appendix C. We thank Andrea Antinucci for comments on this point and for careful explanation of his recent paper [82].

### 4.4.1 Explicit breaking via gauging and mixed anomalies

First, it is often the case that we can break a symmetry by gauging a *different* symmetry with which it has a mixed anomaly.[15] One can attempt this strategy for solving the 2$d$ Maxwell theory analogue of the Strong CP problem, as follows. A well known fact about Maxwell theory in any number of dimensions is that it has a $U(1)_e^{(1)} \times U(1)_m^{(d-3)}$ symmetry under which Wilson lines and 't Hooft operators are charged. There is an obstruction to gauging these two symmetries at the same time, or a mixed 't Hooft anomaly, that can be succinctly encapsulated in terms of the background gauge fields $B_e$ and $B_m$ by its anomaly polynomial,

$$\mathcal{A} = \frac{1}{2\pi}dB_e \wedge dB_m\,. \tag{47}$$

In the 2$d$ case these facts also hold, with the background field for the magnetic symmetry being $\theta$ itself. In this case the anomaly polynomial is,

$$\mathcal{A}^{2d} = \frac{1}{2\pi}dB_e \wedge d\theta\,. \tag{48}$$

It follows that an easy way of breaking the magnetic $(-1)$-form symmetry is to gauge the electric symmetry. The gauging is implemented by coupling the electric symmetry to a background gauge field $B_e$, adding suitable local counterterms and summing over the background field configurations in the path integral. The resulting action is[16]

$$S = \int -\frac{1}{2e^2}(F - B_e) \wedge \star(F - B_e) + \frac{1}{2\pi}\theta(F - B_e)\,. \tag{49}$$

A first observation is that the kinetic term for the gauge field becomes a Stueckelberg-like coupling and the $U(1)$ gauge field is "eaten" by $B_e$. This mass term removes the pole from the photon 2-point function, destroying the long range force. Thus, the infrared physics of the theory is trivial, and $\mathcal{X} = 0$. Whether one considers this to be a solution of the 2d CP problem or not is perhaps a matter of semantics: the problem is gone, but so is all of the physics, since the photon is gapped.

Even this pyrrhic victory is lost in the case of the actual Strong CP problem for QCD. The analogue would be to gauge a $U(1)$ 3-form symmetry that has an anomaly polynomial of the form

$$\mathcal{A}^{4d} = \frac{1}{2\pi}dB_4 \wedge d\theta\,, \tag{50}$$

with $B_4$ a background gauge field for the 3-form symmetry. However, QCD has no such 3-form symmetry! The 3-form gauge field associated with the Kogut-Susskind pole emerges in the IR, rather than existing as a fundamental UV field. There is no electric 3-form global symmetry associated with it that we can gauge.

### 4.4.2 Explicit breaking in the UV

It is possible that the $(-1)$-form $U(1)$ global symmetry associated with QCD instantons is explicitly broken in the UV. Because the symmetry is topological, we expect that this will occur only when the gauge group itself is somehow modified in the UV. An example was discussed in [19]. Suppose that the Standard Model gauge group is embedded in SU(5) (and let us

---

[15]There are many exceptions to this statement when gauging gives rise to a 2-group structure or non-invertible symmetries, see for instance [83–87]. Here we restrict to anomalies that break the ungauged symmetry.

[16]A similar computation has recently appeared in section 4.1.2 of [88].

ignore fermions for the moment, which slightly complicate the story without changing the punchline). The UV theory has a single $(-1)$-form Chern-Weil symmetry with current

$$j_{0\,\text{SU(5)}} = \frac{1}{8\pi^2} \star \text{tr}(F_{\text{SU(5)}} \times F_{\text{SU(5)}}). \tag{51}$$

In the IR however, there are three such Chern-Weil symmetries, one for each field strength. Clearly, two linear combinations thereof are emergent in the low energy theory, and explicitly broken in the UV. One could then ask: could the UV breaking of an emergent $(-1)$-form symmetry be sufficient to remove the Nambu-Goldstone pole and solve the Strong CP problem? From one point of view, the answer should be no, as it would be a dramatic failure of decoupling if physics at the GUT scale could set the topological susceptibility computed in the infrared limit of QCD to zero. From a different point of view, however, one may have the intuition that Nambu-Goldstone poles are fragile and easily removed by UV effects. Thus, it is worth discussing this point in more detail.

A well-known fact about Nambu-Goldstone bosons parametrizing the degenerate vacua of a spontaneously broken 0-form symmetry is that, if the symmetry is not exact in the UV, they get a small mass [89]. Consider for instance a symmetry that is explicitly broken by some irrelevant coupling with a characteristic scale $\Lambda_{\text{UV}}$. If the emergent symmetry is spontaneously broken at a scale $\Lambda_{\text{IR}}$, the pseudo-Goldstone boson will typically have a mass scaling like $\sim (\Lambda_{\text{IR}}/\Lambda_{\text{UV}})^p$, where $p$ is some power depending on the specific details.

A surprising feature of Nambu-Goldstone bosons for spontaneously broken 1-form symmetries is that their masslessness remains protected even if the symmetry is only approximate in the sense above. In other words, 1-form symmetries (and higher) are exact emergent symmetries [56]. An example of this fact is the electromagnetic field that we observe in nature. At low energies there are two U(1) symmetries, electric and magnetic. At high energies these two symmetries are explicitly broken by the presence of electric fermions and, presumably, magnetic monopoles. Those two symmetries are spontaneously broken and the Nambu-Goldstone boson is the photon, which is exactly massless despite the explicit breaking in the UV. More detailed examples were given in [56]. A difference with 0-form symmetries is that no local operator can be charged under a 1-form symmetry, which implies that emergent 1-form symmetries are exact in perturbation theory. The standard lore is that this helps in keeping the photon massless.

For the case of $(-1)$-form symmetries one can pose a similar question: If the $(-1)$-form symmetry is emergent in the IR, is the massless nature of the emergent gauge field, i.e., the pole, protected? In other words, are emergent $(-1)$-form symmetries exact? This question is relevant because the pole is behind all the features of spontaneously broken $(-1)$-form symmetries and, in particular, if there is no pole, there is no vacuum electric field and, thus, no Strong CP problem. If $(-1)$-form symmetries behave like 0-form symmetries it should be possible to lift the pole by changing the UV physics in such a way that the $(-1)$-form symmetry is explicitly broken.

Again, it is useful to consider the case of Maxwell theory in various dimensions. In $3d$, where the photon is dual to a compact scalar, Maxwell theory has an electric 1-form symmetry and a magnetic 0-form symmetry. The magnetic 0-form symmetry can be explicitly broken in the UV by embedding U(1) gauge theory in SU(2) gauge theory, higgsed by a real adjoint scalar field. This theory admits a famous semiclassical analysis of confinement due to Polyakov [59], in which magnetic monopoles (which are instantons in $3d$) produce an exponentially small mass for the dual photon. Thus, in this case, the would-be Nambu-Goldstone boson is removed by the explicit breaking of a 0-form symmetry.

On the other hand, in $2d$ Maxwell theory, the magnetic symmetry is a $(-1)$-form symmetry. Again, the magnetic symmetry can be explicitly broken by embedding the U(1) gauge theory in SU(2); no axion coupling that UV completes $\frac{1}{2\pi}\theta F$ is possible in that theory, because the

SU(2) field strength is not gauge invariant. There are no instanton effects in this theory, and the photon should remain massless. Thus, in this case we expect that the pole associated with the spontaneous breaking of the $(-1)$-form symmetry is still present in the IR, despite the explicit breaking of the symmetry in the UV. We expect a similar behavior in the case of SU(5) breaking into the SM gauge group in $4d$, and leave a detailed investigation for a future study.

We expect that the lesson here generalizes: a Nambu-Goldstone pole can be removed only in the case where the Nambu-Goldstone is protected by a 0-form symmetry that is explicitly broken in the UV. The Kogut-Susskind pole is associated with an emergent $(d-1)$-form gauge field, and can be protected by either a $(d-1)$-form symmetry or a $(-1)$-form symmetry, and hence its existence is robust against UV symmetry breaking in $d > 1$ spacetime dimensions. Thus, embedding the Standard Model in a GUT should not have any impact on the Strong CP problem.

### 4.5 Solving the problem with gauged reflection symmetries

Aside from the axion, the most well-studied solution to the Strong CP problem assumes a fundamental spacetime reflection symmetry, which is either a generalized parity symmetry [90] or CP symmetry [91–94]. Here we will refer to the latter case, generally known as Nelson-Barr models, though our remarks will apply more broadly. Theories with a spacetime reflection symmetry can be defined on non-orientable manifolds. On such a space, $\frac{1}{8\pi^2}\mathrm{tr}(F \wedge F)$ is not defined, because $F \wedge F$ is an ordinary differential form, but only pseudo-forms can be integrated without a choice of orientation. Thus, the instanton number is not well-defined (although a topological invariant valued in $\mathbb{Z}_2$ survives). However, by our definition, these theories still have a $(-1)$-form U(1) global symmetry, because they can be consistently coupled to a background *pseudoscalar* axion field $\theta(x)$, which transforms with a minus sign under spacetime reflections. Furthermore, the symmetry is still spontaneously broken, because *on Minkowski space* we can still turn on an arbitrary constant $\bar{\theta}$ term and evaluate a nonzero topological susceptibility (31). However, the only constant $\theta(x)$ backgrounds that can be defined on an *arbitrary space* are $\bar{\theta} = 0$ and $\bar{\theta} = \pi$. Thus, the reflection symmetry requires that the theory be defined with one of these two special $\bar{\theta}$ terms, and the Strong CP problem could, in principle, be solved.

The difficulty begins when we recall that the world in which we live is not CP symmetric, and indeed the CP-violating phase in the CKM matrix is an $O(1)$ number. Thus, if we live in a universe with an underlying CP symmetry, the symmetry must be spontaneously broken, and (at least as measured by the CKM phase) badly so. Below the scale of CP breaking, we should match the fundamental theory onto a theory without CP, and such a theory in principle admits an arbitrary constant $\bar{\theta}$ term. The low-energy value of $\bar{\theta}$ need not be one of the special values $\bar{\theta} = 0$ or $\pi$ defining the theory in the ultraviolet, because integrating out massive particles that couple to CP-breaking can generate effective contributions to $\bar{\theta}$ in the IR. Nelson-Barr models are engineered so that such effects are small, whereas the CKM phase is large. It is difficult to give a purely symmetry-based explanation of how they work, without delving into the detailed structure of the quark mass matrices, which must be enforced with additional (model-dependent) gauge symmetries.

## 5 Outlook

In this work we have extended the notion of spontaneous breaking to $(-1)$-form U(1) symmetries and started the exploration of its applications. We finish this text with some open questions.

- We have provided a useful working definition of a $(-1)$-form U(1) symmetry and its spontaneous breaking, but it would be useful to put $(-1)$-form symmetries in general on a more similar footing to other $p$-form symmetries in QFT. For example, the SymTFT approach could be a useful way to formulate $(-1)$-form symmetries. It would also be interesting to gain a better understanding of whether $(-1)$-form $\mathbb{R}$ symmetries are a useful concept in QFT.

- We have explored several solutions to the Strong CP problem which, broadly speaking, aim at removing the Nambu-Goldstone *field* by either gauging or explicitly breaking the $(-1)$-form symmetry. It turns out that gauging has been extensively covered in the literature. On the other hand, it seems to us that explicit breaking is still poorly understood and we hope to study it further in future work. Besides, the explicit breaking of the $(-1)$-form symmetry by monopoles has not been thoroughly investigated except in the case of U(1) gauge theory [19]. It would be interesting to examine the case of more general gauge groups, including Grand Unified Theories.

- A fundamental ingredient in our understanding of spontaneous breaking of continuous (higher form) symmetries is the Goldstone Theorem. While we have established the presence of a pole in the 2-point function of a Nambu-Goldstone *field* whenever a $(-1)$-form U(1) symmetry is spontaneously broken, we have not been able to explicitly prove a theorem that follows the reasoning of the usual Goldstone theorem. Difficulties arise because the symmetry operator fills the entire spacetime and cannot be deformed, and because there are no $(-1)$-dimensional charged operators that can obtain vacuum expectation values. A better understanding of the formalism of $(-1)$-form symmetries may overcome these obstacles and provide a more direct Goldstone Theorem.

- While most of the solutions to the Strong CP-problem that are discussed in our paper are concentrated on lifting the $(-1)$-form U(1) symmetry, Nelson-Barr models are conceptually different. Unlike the other solutions that lead to no dependence of the vacuum energy on $\theta$, Nelson-Barr models are constructed such that the value of $\theta$ is small. It remains an open question to understand a generic symmetry-based explanation for such models.

- Axion-like fields play a role in several solutions to longstanding naturalness problems in particle physics. A prime example is the hierarchy problem, which sees potential mitigation through the relaxion model [95,96]. This model introduces an axion-like field having a coupling with the Higgs mass term. Another example is the axion monodromy framework for inflation [45]. In these models the axion couplings appear to violate the axion periodicity but it is restored by monodromy, much as in the 2$d$ Maxwell example we discussed in Sec. 2.3. Understanding the interplay between monodromy and SSB of $(-1)$-form symmetries in such models would be interesting.

We hope that this novel case of spontaneous symmetry breaking will prove a useful unifying tool for physical phenomena.

# Acknowledgements

We wish to thank Andrea Antinucci, Riccardo Argurio, Diego Delmastro, Ben Heidenreich, Ohad Mamroud, Jake McNamara, Miguel Montero, Tom Rudelius, John Stout, Ho Tat Lam, Angel Uranga, and Irene Valenzuela for insightful conversations.

EGV also thanks the Simons Center for Geometry and Physics and its Summer Physics Workshop for kind hospitality where part of this work was completed. MS would like to thank Fermilab theory division for their hospitality.

**Funding information** The work of DA is supported by the U.S. Department of Energy (DOE) under Award DE-SC0015845. The work of EGV has been partially supported by Margarita Salas award CA1/RSUE/2021-00738, by MIUR PRIN Grant 2020KR4KN2 "String Theory as a bridge between Gauge Theories and Quantum Gravity" and by INFN Iniziativa Specifica ST&FI. MR is supported by the DOE Grant DE-SC0013607. MS is supported by JSPS KAKENHI Grant Numbers JP22J00537.

# A   The 2$d$ Abelian-Higgs model

A nice exposition of this model, which we follow, may be found in [68, 97]. Consider the action,

$$S = \int d^2x \frac{1}{2e^2}F_{01}^2 + \frac{\theta}{2\pi}F_{01} + |D\phi|^2 - m^2|\phi|^2 - \frac{\lambda}{2}|\phi|^4. \tag{A.1}$$

As already mentioned, in 2$d$ the gauge coupling is dimensionful and the theory is strongly coupled in the IR. Hence, the regime $|m^2| \lesssim e^2$ will be complicated to solve. Consider instead $|m^2| \gg e^2$. There are then two regimes to consider,

- **$m^2 \gg e^2$**: In this case the gauge symmetry is not spontaneously broken and the theory is just electrodynamics with a heavy scalar meson. The behaviour is similar to the massive Schwinger model. In particular, there is a vacuum electric field, a non-zero topological susceptibility and a long range constant force mediated by the photon. We conclude that there is a magnetic $(-1)$-form symmetry in the IR which is spontaneously broken.

- **$m^2 \ll e^2$**: In this case the gauge symmetry is spontaneously broken by the condensation of the scalar field. The naive expectation is that the photon becomes massive, the long-range force is screened, the topological susceptibility vanishes and there is no electric field in the vacuum. We therefore expect that the $(-1)$-form symmetry is not spontaneously broken. It turns out that this expectation is wrong. Due to non-perturbative effects mediated by instantons (which are vortices in 2 dimensions), the gauge symmetry is restored, there is a long-range force between probe particles and there is a vacuum electric field which depends linearly with $\theta$. In fact the physics is the same as in the $m^2 \gg e^2$ regime but all effects are exponentially suppressed. We learn that the $(-1)$-form symmetry is, contrary to expectation, realized in the IR and spontaneously broken!

# B   The $CP^{N-1}$ model

This model has been extensively discussed in the literature, we follow [68, 98]. The $CP^{N-1}$ model can be defined by the following Euclidean action,

$$S = \int d^2x \frac{1}{2e^2}|F|^2 + i\frac{\theta}{2\pi}F + \sum_{a=1}^{N}|\mathcal{D}\phi_a|^2 + \frac{\lambda}{2}\left(\sum_{a=1}^{N}|\phi_a|^2 - v^2\right)^2, \tag{B.1}$$

which describes a set of $N$ massive complex scalar fields with an SU($N$) global symmetry and coupled to a U(1) gauge field. Classically we expect the scalar potential to be minimized,

spontaneously breaking the SU($N$) symmetry to SU($N-1$) × U(1),

$$|\phi_a|^2 = v^2 \,. \tag{B.2}$$

Thus, classically, the low energy is described by $N-1$ massless scalar fields (Goldstone bosons) with target space,

$$\mathbf{CP}^{N-1} = \frac{\mathrm{SU}(N)}{\mathrm{SU}(N-1) \times \mathrm{U}(1)} \,. \tag{B.3}$$

This is of course in contradiction with the MWC theorem and it is well known that strong dynamics radically change the low energy physics of this theory. We will not review the computations but merely state the result. It turns out that the low energy dynamics is that of $N$ massive scalar fields coupled to a U(1) gauge field, whose dynamics is emergent in the IR.[17] The mass is given by a dynamically generated scale $\Lambda_{CP^{N-1}}$ analog to $\Lambda_{QCD}$. The resulting dynamics are pretty much the same as the ones of the abelian-Higgs model in the unbroken phase:

- There is an electric field in the vacuum. In the large N limit it was computed in [38,99],

$$\langle F_{01} \rangle \sim \frac{\Lambda_{CP^{N-1}}^2}{N} \theta + O(1/N) \,. \tag{B.4}$$

- It follows that the topological susceptibility, which is the order parameter for the $(-1)$-form symmetry SSB, takes a non-zero value,

$$\mathcal{X} \sim \frac{\Lambda_{CP^{N-1}}^2}{N} \,. \tag{B.5}$$

- There is a long range force between fractionally quantized probe particles. Integer quantized particles don't experience such force because they are screened by Schwinger pair production.

- The spectrum is composed of mesons. As the $\theta$ angle is dialed from 0 to $2\pi$ a $\phi\phi^\star$ pair is created and the spectrum undergoes a flow.

- Interestingly all these phenomena are not exponentially suppressed as befits an instanton effect, signaling that one can't hope to explain them using semiclassical techniques.

- We conclude that there is a $(-1)$-form symmetry which is spontaneously broken and all the expected features are present.

Note that this model has many of the salient features of QCD. It is well-known that it has $\theta$-vacua, a dynamically generated scale and instantons that are insufficient to explain the low energy physics. We learn now that it shares a further feature with QCD, namely an $(-1)$-form symmetry which is spontaneously broken.

## C   A SymTFT for 2$d$ Maxwell theory.

The SymTFT [100–102] of a given $d$ dimensional Quantum Field Theory $\mathcal{T}_d$ is a $d+1$ TQFT that encodes the (categorical) symmetry of $\mathcal{T}_d$ and of all other theories that can be obtained from $\mathcal{T}_d$ by a topological manipulation. The SymTFT is placed on a $(d+1)$ slab with two boundaries. On

---

[17]In fact, if one starts with no kinetic term for the UV gauge field a nonzero kinetic term is dynamically generated in the IR. In this sense, the U(1) dynamics are emergent.

one of them the boundary condition is not topological and the local degrees of freedom live. On the other boundary a topological boundary condition is imposed that prescribes the symmetry of $\mathcal{T}_d$ once the slab is collapsed to recover $\mathcal{T}_d$. Different topological boundary conditions encode the symmetry of theories obtained from $\mathcal{T}_d$ by a topological manipulation.

This construction has recently been extended to abelian continuous symmetries in [82, 103]. In this appendix we present the SymTFT for $2d$ abelian gauge theories, which is an application of [82]. This construction puts $(-1)$-form $U(1)$ symmetries in the same footing as more familiar symmetries and also clarifies the relation between the $(-1)$-form symmetries of abelian gauge theories with different global forms of the gauge group.

Consider a $3d$ BF theory with action,

$$S = \frac{1}{2\pi} \int \phi \, \mathrm{d}b_2 \,, \tag{C.1}$$

where both $\phi$ and $b_2$ are $\mathbb{R}$ gauge fields.[18] This means that they don't have any large gauge transformations and charged operators with arbitrary real coefficients are allowed. In this case there are local operators and surfaces,

$$U_\alpha(x) = \mathrm{e}^{i\alpha\phi(x)} \,, \qquad \tilde{U}_\beta(\Sigma_2) = \mathrm{e}^{i\beta \int_{\Sigma_2} b_2} \,, \tag{C.2}$$

where $\alpha, \beta$ are real-valued. The braiding between these operators is,

$$\langle U_\alpha(x)\tilde{U}_\beta(\Sigma_2)\rangle = e^{2\pi\alpha\beta\cdot\mathrm{Link}(x,\Sigma_2)} \,. \tag{C.3}$$

This SymTFT encodes the symmetry of continuous free abelian gauge theories in $2d$. Different topological boundary conditions correspond to a choice of mutually transparent bulk operators that can terminate on the boundary, i.e. their braiding is trivial. Different boundary conditions give different symmetries on the boundary. Here we mention those corresponding to the global forms that we have encountered in the main text.[19]

- Dirichlet Boundary Conditions (DBC's) for $b_2$ allow the Wilson surfaces $\tilde{U}_\beta(\Sigma_2) = \mathrm{e}^{i\beta \oint_{\Sigma_2} b_2}$ to end on the boundary, giving rise to a $\mathbb{R}$ 1-form symmetry. The topological symmetry operators on the boundary are $U_\alpha(x)$. For the variational problem to be well posed $\phi$ must obey Neumann Boundary Conditions (NBC's), which imply that it must be summed over in the $2d$ theory. This sum explicitly implements the topological gauging mentioned in footnote 14. The total symmetry of the $2d$ theory is an $\mathbb{R}$ 1-form symmetry, which is the symmetry of the $\mathbb{R}$ $2d$ gauge theory that we met in section 4.3.

- Mixed boundary conditions are allowed. In particular, one may impose NBC's for the $\mathbb{Z}$ piece of both fields and DBC's for the remaining $U(1) \simeq \mathbb{R}/\mathbb{Z}$ pieces [103]. These boundary conditions allow operators $\tilde{U}_\beta(\Sigma_2)$ with $\beta \in \mathbb{Z}$ to end on the boundary. The endpoints become the charged lines under a $U(1)^{(1)}$ symmetry. The remaining operators $\tilde{U}_\beta(\Sigma_2)$ with $\beta \in U(1)$ can be placed on the boundary and correspond to symmetry operators generating a $U(1)^{(-1)}$ symmetry. The operators $U_\alpha(x) = \mathrm{e}^{i\alpha\phi(x)}$, $\alpha \in \mathbb{Z}$ *can't end* on the boundary because they are zero-dimensional, so the $U(1)^{(-1)}$ symmetry *does not have charged operators*. Finally, the operators $U_\alpha(x) = \mathrm{e}^{i\alpha\phi(x)}$, $\alpha \in U(1)$ are topological on the boundary and generate the $U(1)^{(1)}$ symmetry. We conclude that the total symmetry is,

$$U(1)^{(1)} \times U(1)^{(-1)} \,, \tag{C.4}$$

which matches the symmetry of the $U(1)$ gauge theory discussed in section 2.3.

---

[18]For the case of $\phi$ this means that it is a non-compact scalar field.

[19]The different boundary conditions can be expressed in terms of a variational problem that must be well-defined, see [87,103] for a similar discussion. We refrain from going into such details and merely state the results here.

The SymTFT of the gauge theories with electric matter can be similarly realized by turning $\phi$ into a compact scalar. Through this exercise we see that $(-1)$-form symmetries are very similar to more usual symmetries, at least from the SymTFT point of view. We plan to return to these considerations in more generality and depth in future work.

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
