# Peer review of "Spontaneously Broken $\bm{(-1)}$-Form U(1) Symmetries"

_SciPost Physics, doi:SciPost Phys. 17, 031 (2024)_

## Round 1 · Referee Report · Anonymous (Referee 1) · 2024-5-15

Strengths
The authors provide a new perspective on the theta angle and the strong CP problem in the language of (-1)-form symmetries. The instanton number density is viewed as a local Noether current for the (-1)-form symmetry, and the axion field is the gauge field. They define the spontaneous breaking of a (-1)-form symmetry in terms of a nonzero topological susceptibility. The discussion in 2d QED and the Schwinger model is particularly illuminating. The new perspective provides a reinterpretation of existing proposals to solve the strong CP problems. It also leads to some new attempts, but unfortunately with no success.
Weaknesses
The (-1)-form symmetry is notoriously subtle, and it is not clear in what sense it can be viewed as a generalized symmetry. As the authors acknowledged, not every aspect of the higher-form symmetry has a counterpart for the (-1)-form symmetry. In some discussions, such as the one for the spontaneous symmetry breaking, the authors acknowledged that many usual definitions do not apply for the (-1)-form symmetry, and they focused on one that works. Because of this, I find the argument less convincing than those for the higher-form symmetries.
Report
The authors provide an interesting new perspective on the strong CP problem and the theta angle. While it is still early to say if this perspective will lead to groundbreaking new insights, I consider this a worthwhile approach. I recommend the paper for publication with minor revisions.
Requested changes
1. In (2.11), F(X) should be F(x).
2. Above (2.13), the authors should say that the Dirac fermion has unit gauge charge. Otherwise the model has a 1-form symmetry.
3. In (2.14), the second term suggests that ϕ is dimensionless, but then there should be a dimensionful coupling constant in the third term.
4. Similar dimension issue appears in (2.18). Also, the ϵ in the last term appears to be a typo.
5. I am confused by the relation between S in (2.13) and S' in (2.14). Below (2.13), the authors say S has a (-1)-form global symmetry. But in (2.14) it's clear that the (-1)-form symmetry is gauged. If S and S' are equivalent QFT, how come the former has a (-1)-form global symmetry, while the latter has a (-1)-form gauge symmetry? Note that usually bosonization is not an equivalence between two QFTs; rather it means that one can be obtained from the other by gauging the fermion parity. However, for the Schwinger model, since the fermion parity is part of the gauge group, (2.13) and (2.14) are equivalent globally. Thus I am confused why the (-1)-form global symmetry does not match between (2.13) and (2.14).
Recommendation
Ask for minor revision
Author: Eduardo Garcia Valdecasas on 2024-06-07 [id 4546]
(in reply to Report 1 on 2024-05-15)
Thanks for the careful report. We have addressed the requested changes in the following way.
- We have changed F(X) to F(x).
- We have specified the charge of the Dirac fermion above eq. 2.13.
- The referee asked about the dimensions of the compact boson phi. Indeed, it is dimensionless, but it is important to fix its normalization. We have assumed it to have period 2pi, which then requires us to fix the coefficient in front of its kinetic term. This constant is fixed by the bosonization dictionary for a boson that is dual to a free Dirac fermion. We have added footnote 10 clarifying this fact.
- As the referee noted, we introduced \epsilon without explaining its meaning. It is a UV regulator that is needed when bosonizing massive fermions. We have added an explanation below eq. 2.18.
- The referee is correct in pointing out that the symmetry of 2.13 is the same as the one as 2.14, including the (-1)-form symmetry. This is indeed the point of that discussion. We have modified the paragraph below eq. 2.13 to anticipate the fact that the (-1)-form symmetry is gauged in the Schwinger model. We hope this is enough to make the point clear. We have also added footnote 9 with the comment about fermion parity, which we have found insightful.
Author: Eduardo Garcia Valdecasas on 2024-06-07 [id 4548]
(in reply to Report 3 on 2024-06-03)We thank the referee for the positive report.
The referee ask an interesting question regarding axions with potentials other than the one arising from QCD. We have added a paragraph at the end of section 4.1. explaining this case and its connection with the axion quality problem.

---

## Round 1 · Referee Report · Anonymous (Referee 2) · 2024-5-22

Strengths
This is a timely, reasonably well-written article discussing continuous (-1)-form symmetries, interpreted via their potentials (as theta angles), and in particular spontaneous symmetry breaking in this context. An application to the strong CP problem is discussed.
Weaknesses
These symmetries are rather subtle to work with, as the other referee observed.
Report
It's reasonably well-written and timely, so I recommend it for publication.
Requested changes
No requested changes beyond those noted by the other referee.
Recommendation
Publish (meets expectations and criteria for this Journal)
Author: Eduardo Garcia Valdecasas on 2024-06-07 [id 4547]
(in reply to Report 2 on 2024-05-22)We thank the referee for the report. We have implemented the changes of referee 1.

---

## Round 1 · Referee Report · Anonymous (Referee 3) · 2024-6-3

Report
This work extended the notion of spontaneous symmetry breaking to -1 form symmetry and proposed a concrete criterion to diagnose symmetry breaking based on the topological susceptibility. It then used this new perspective to unify various solutions to the Strong CP problem.
This paper furthered our understanding of generalized symmetries and provided an interesting connection between this subject and particle phenomenology. I recommend the publication of this manuscript in SciPost.
One question related to the interpretation of axion solution to the Strong CP problem as gauging. Currently, in the paper, it assumed that the axion has no potential when it couples to QCD. If there is a potential added by hand or generated from other sources before coupling to QCD, is it still possible to interpret axion solution as gauging? It would be nice to add a few words about this in the paper although it is not necessary.
Recommendation
Publish (easily meets expectations and criteria for this Journal; among top 50%)

---

## Round 2 · Referee Report · Anonymous (Referee 1) · 2024-6-7

Report

The authors have addressed most of my comments but I am still confused by item 5 on the List of Changes. In the paragraph below (2.13), on the one hand, the authors state that " The symmetry of this theory is just U(1)_m^(−1)". On the other hand, in the same paragraph, they state "The explanation will be that the U(1)^(−1)_m symmetry has been gauged." I would kindly like to ask the authors to clarify if the (-1)-form symmetry is a global symmetry or a gauge symmetry of this theory.

Recommendation

Ask for minor revision

  • validity: -
  • significance: -
  • originality: -
  • clarity: -
  • formatting: -
  • grammar: -

Author:  Eduardo Garcia Valdecasas  on 2024-06-11  [id 4557]

(in reply to Report 2 on 2024-06-07)

The U(1)^(−1)_m symmetry is gauged. We have modified the paragraph below eq. 2.13 to be more clear about this fact.

---

## Round 2 · Referee Report · Anonymous (Referee 3) · 2024-6-7

Report

The authors have addressed my comments. I therefore recommend the direct publication of this manuscript in SciPost.

Recommendation

Publish (easily meets expectations and criteria for this Journal; among top 50%)

---

## Round 2 · Author Response

We have addressed the referees suggestions.

---

## Round 2 · List of Changes

1. We have changed F(X) to F(x).
  2. We have specified the charge of the Dirac fermion above eq. 2.13.
  3. The referee 1 asked about the dimensions of the compact boson phi. Indeed, it is dimensionless, but it is important to fix its normalization. We have assumed it to have period 2pi, which then requires us to fix the coefficient in front of its kinetic term. This constant is fixed by the bosonization dictionary for a boson that is dual to a free Dirac fermion. We have added footnote 10 clarifying this fact.
  4. As the referee 1 noted, we introduced \epsilon without explaining its meaning. It is a UV regulator that is needed when bosonizing massive fermions. We have added an explanation below eq. 2.18.
  5. The referee 1 is correct in pointing out that the symmetry of 2.13 is the same as the one as 2.14, including the (-1)-form symmetry. This is indeed the point of that discussion. We have modified the paragraph below eq. 2.13 to anticipate the fact that the (-1)-form symmetry is gauged in the Schwinger model. We hope this is enough to make the point clear. We have also added footnote 9 with the comment about fermion parity, which we have found insightful.
  6. The referee 3 ask an interesting question regarding axions with potentials other than the one arising from QCD. We have added a paragraph at the end of section 4.1. explaining this case and its connection with the axion quality problem.

---

## Round 3 · Referee Report · Anonymous (Referee 1) · 2024-6-11

Report

The authors have addressed all my comments and the manuscript is ready for publication in my opinion.

Recommendation

Publish (easily meets expectations and criteria for this Journal; among top 50%)

---

## Round 3 · List of Changes

We have modified the paragraph below eq. 2.13 to further clarify that the (-1)-form symmetry is gauged in the Schwinger model.

---

## Editorial Decision

published